# Attending Informal Preschools and Daycare Centers Is a Risk Factor for Underweight, Stunting and Wasting in Children under the Age of Five Years in Underprivileged Communities in South Africa

**DOI:** 10.3390/ijerph16142589

**Published:** 2019-07-20

**Authors:** Sphiwe Madiba, Paul Kiprono Chelule, Mathildah Mpata Mokgatle

**Affiliations:** Department of Public Health, Sefako Makgatho Health Sciences University, Pretoria 0001, South Africa

**Keywords:** children under five years, undernutrition, preschool, daycare centers, South Africa, informal settlements

## Abstract

The study objectives were to determine the nutritional status of children between the ages of 12–60 months and to establish the association between attending preschool and the prevalence of undernutrition. This was a cross-sectional survey conducted in health facilities in Tshwane district in South Africa, consisting of both a questionnaire and anthropometric measures of 1256 mothers and their children. Weight-for-age (WAZ), height-for age (HAZ) and BMI-for-age (BAZ) were calculated and bivariate and multivariable analysis was performed to establish association. The results showed that child-related factors, namely birthweight, age, gender, and attending preschool increased the risk of undernutrition. Children over the age of 24 months were likely to be stunted and underweight. Maternal education reduced the odds of underweight. Children who stayed at home had reduced odds of underweight and stunting. High birthweight reduced the odds of wasting and underweight. The risks for undernutrition are multifaceted, but children who attend preschool have an increased risk of undernutrition. The risk of undernutrition increased with age and coincided with the time of cessation of breast-feeding and attendance at daycare or preschool. The complementary role of quality childcare in preschools and daycare centers is vital in alleviating the problem of undernutrition in underprivileged communities.

## 1. Introduction

Child undernutrition continues to be a major global public health problem that is responsible for lost potential, morbidity, and death. In sub-Saharan Africa, child undernutrition is a huge public health problem that is not properly given the priority that it deserves [1,2,3]. Governments of some of the countries with the highest levels of undernutrition are faced with multiple challenges and undernutrition often does not feature prominently among these problems, unless it becomes very severe and widespread [4]. This could be because undernutrition is a complex phenomenon that has multiple causes. The consequence is that undernutrition is responsible for nearly half of all deaths in children under five years of age and responsible for around 3.1 million deaths in this population annually [5].

Undernutrition is the outcome of insufficient food intake and repeated infectious diseases [3]. Low weight for age or underweight, low height for age or stunting, and low weight for height or wasting are the three main indicators used to define undernutrition [6]. Compared to other forms of undernutrition, stunting remains a problem of greater magnitude than underweight or wasting among children under five years old in the developing world. Stunting accurately reflects the cumulative effects of undernutrition that occur even before birth and during the most critical periods for growth and development and is aggravated by illness. Most countries have stunting rates that are much higher than their underweight rates, for example, globally, an estimated one-third (195 million) children are stunted, whereas 129 million are underweight [3,4]. In sub-Saharan Africa, 40% of children under five years of age are stunted, 21% are underweight, and 9% are wasted [7,8]. Childhood stunting results in delayed mental development and school performance [3,7].

Nearly one in four children under the age of five years in the developing world are underweight [9]. Underweight is a composite indicator, because a child who is underweight can also be stunted or wasted or both. Thus, the mortality risk of children who are even mildly underweight is increased, and severely underweight children are at even greater risk [3]. Children who are underweight are at greater risk of contracting pneumonia, measles, diarrhea, malaria, and HIV/AIDS, and of dying from these conditions [10]. While wasting refers to low weight-for-height and is a symptom of acute undernutrition, it is a consequence of insufficient food intake or a high incidence of infectious diseases, especially diarrhea (World Health Organization (WHO), 2010).

The factors influencing child undernutrition are multifaceted but the underlying causes are; inadequate access to food or food security, inadequate care for children and women, insufficient health services, and unhealthy environment [7]. Food insecurity can be defined as the inability to access adequate quantities of nutritious foods required for optimal growth and development [2]. As such, household food insecurity is assumed to affect the nutritional status of children by compromising quantity and quality of dietary intake. Although food insecurity has been identified as one of the underlying causes of undernutrition, food security at the household and individual level is a necessary but not sufficient condition for adequate nutrition [11]. The nutritional status of children can be affected by other factors such as low parental education, poor feeding practices, maternal nutritional status, number of under five children in one family, poverty, access to health services, as well as urban and rural differences [2,3].

In 1996, the South African National Department of Health (NDoH) introduced the Integrated Nutrition Programme (INP) to improve the nutritional status of pregnant women and children under the age of five years. However, the persistent high rates of stunting indicating chronic undernutrition suggests that the implementation of the INP program has not reduced the prevalence of early life malnutrition. According to the latest South African National Health and Nutrition Examination Survey (SANHANES) data (2012), 21% of South African children under the age of five years are stunted. The The United Nations Children’s Fund (UNICEF) includes South Africa as one of 24 high-burden countries that account for 80% of the world’s stunted children [4].

One of the limitations of intervention strategies to reduce child undernutrition is the focus on household food security, without taking into consideration the complementary role of quality childcare [12]. Childcare as a concept has been identified as a major factor influencing the nutritional status of children under the age of five years [11,12,13,14,15]. The childcare concept is outline in a conceptual model by the UNICEF which indicates that inadequate infant and young child feeding practices comprise a chief underlying cause of child undernutrition [16]. Nutritional interventions assume that children under the age of five years are cared for by maternal caregivers within the household. However, because mothers take up employment, a considerable number of children are cared for in daycare centers and preschool facilities at a young age by non-maternal caregivers [14,17]. Of public health significance is the link between children cared for at daycare or in preschool centers and the greater risk of acquiring infections [13,17,18]. Since undernutrition is aggravated by diseases [14,19], it could be concluded that many children who spend most of their day at day centers or preschools are at high risk for both malnutrition and acquiring infections. Moreover, daycare centers in most poor resourced communities in South Africa are often informal [20] and are not addressed by nutritional interventions. In South Africa, the National School Nutrition Programme responds to nutritional needs of children at school and at registered early learning centers.

This study was motivated by the diversity of population groups accessing primary health care facilities and the co-existence of urban, peri-urban, and informal settlements or shacks within the same health district. Although there is evidence of differences in urban-rural malnutrition in most sub-Saharan countries [21,22], this gap is narrowing due to an increase in urban malnutrition and the increased transition from rural homes to the generally poor living conditions in urban informal settlements or shacks [22]. Furthermore, the nutritional status of children under the age of five years old varies significantly among the nine provinces and within each province in South Africa. In addition, there is a dearth of data on the influence of care practices on child undernutrition, taking into consideration the context of children at home and attending day care centers and preschools in poor resourced communities. Thus, the purpose of this study was to determine the nutritional status of children between the ages of 12–60 months and related factors and to establish the relationship between attending daycare or preschool and the prevalence of undernutrition. This research was undertaken in light of increasing early utilization of daycare and preschool facilities by working mothers and the promotion of early childhood learning. The results from this study will highlight the need to focus interventions to respond to the nutritional needs of children under the age of five years attending informal daycare and preschool centers in under privileged communities.

## 2. Methodology

### 2.1. Study Design

This was a health facility based cross-sectional survey which consisted of both a questionnaire and anthropometric measures. The study was conducted from June to December 2014 involving mothers with children aged 0–60 months in 18 health facilities in one of the health districts in Gauteng Province, South Africa. The district’s health facilities serve diverse population groups from urban, peri-urban, and informal settlement areas. We included only children accompanied by their biological mothers who could provide accurate information about their children. The mothers were recruited from the well-baby clinics as they brought their children for routine immunization.

### 2.2. Sampling

The average number of children who are less than five years of age seen in well-baby clinics in the 18 facilities within the district ranged from *n* = 310 to 370 per month with an average of *n* = 340 per month. With a population size of *n* = 6150, we treated each clinic as a unit. The Raosoft software for sample size calculation at 95% confidence level and 5% margin of error was *n* = 181 participants per clinic. The total sample size in all the 18 clinics was *n* = 3258. At each clinic, the first consenting mother of the under five-year-old child was selected randomly and then every fourth mother was selected using a consecutive sampling technique. For this manuscript, we analyzed data for children aged 12 to 60 months and they accounted for *n* = 1254 of the sample size. Since each clinic was treated as a unit, we collected data from one clinic at a time until the calculated sample size was attained.

### 2.3. Measures

Socio-demographic variables for the mothers included maternal age, marital status, years of education, employment status, occupation, number of live children, family size, household monthly income, household size, access to a tap water source, and sanitation. Child characteristics included the age of the child, the child sex, birth order, childhood illness, attending preschool, birth weight, current height, and current weight. Child feeding practices such as exclusive breast feeding, duration of breast feeding, bottle feeding, and child immunization were considered.

The outcome variable was undernutrition indicated by wasting, stunting and underweight status in children 12–60 months of age. Stunting (height/age), wasting (weight/height) and underweight (weight/age) were defined as per WHO child growth standards. Children with Z scores less than two standard deviations (≤2 SD) below the median of the WHO child growth standards were classified as malnourished (stunted, wasted or underweight), minus 3 SD indicates severe stunting, wasting, and underweight.

### 2.4. Data Collection Procedures

#### 2.4.1. Sociodemographic and Child Characteristics

The study used a structured questionnaire adapted from different standard questionnaires to collect the information from the mothers using Setswana language in order to make the questions clear to the participants. We recruited six field workers fluent in English and Setswana and trained them on the objectives of the study, ethical concerns, and the administration of a standardized tool after the tool was translated to the local language. The field workers were also trained on how to standardize processes for anthropometric measurements, and how to measure the height and weight of the children using WHO recommended procedures [23].

The tool was piloted on 5% of the total sample of the study in one of the health facilities to assess the feasibility and practicality of the data collection procedures. The second author (the supervisor of the fieldwork) checked the questionnaire for completeness on a regular basis throughout the period of data collection.

#### 2.4.2. Child Anthropometric Measures

Anthropometric measurements were carried out using the child’s age, height and weight. Weights of children with light clothing and bare feet were taken using a calibrated portable digital weighing scale (SECA digital scale). The weight of children was taken to the nearest 0.1 kg. The weighing scale was calibrated before weighing each child and checked daily against the standard weight for accuracy.

The length of children less than 24 months (12–24 months) was measured in a lying position and the height for children more than 24 months of age was taken using a length/height board. The height and length of the children were taken to the nearest 0.1 cm. Ethical approval was obtained from the Research Ethics Committee of the University of Limpopo (MREC/H/64/2014). Permission was also obtained from the Research Unit of the Tshwane Health District as well as from the operational managers of the primary health facilities. Before the start of the interviews, written consent was obtained from the participating mothers after the purpose of the study was explained to them. The participation in the study was voluntary for all mothers.

### 2.5. Data Analysis

We used Microsoft Excel^®^ (Microsoft Corpotation, Orlando, FL, USA) and Stata version 13 (StataCorp, College Station, TX, USA) for data entry and analysis respectively. The third author supervised data entry, cleaning, and coding. Descriptive statistics of categorical and dependent variables were presented as frequency distributions and percentages. Anthropometric measurements of weight-for-age (WAZ), height-for age (HAZ) and BMI-for-age (BAZ) were expressed as mean and standard deviations. The Pearson Chi-square test was used to analyze the proportions of categorical nutritional indicators (underweight, stunting, and wasting). Both bivariate and multivariable analysis were performed to establish association among the sociodemographic and other independent variables and nutritional outcome variables (wasting, stunting, and underweight) in children under five years of age. The statistical significance was decided at *p*-value < 0.05. All independent categorical variables with a p value of <0.05 at univariate analysis were included in the multivariable analysis model. In the multivariable analysis, the forward selection method was used to determine the strength of association of categorical variables with outcome variables (stunting, underweight and wasting). Adjusted Odds Ratio with 95% CI was used to determine the strength of association between categorical variables and the nutritional status of the children.

## 3. Results

### 3.1. Mother’s Characteristics

A total of 1256 mothers were included in the survey. Their ages ranged from 15–50 years, the mean age was 29 years (SD = 7.0) and over half were in the age range of 25–35 years. The majority (73%) were unemployed, 67% were recipients of the child support social grant and less than half (45.7%) had high school education (Table 1).

### 3.2. Child Characteristics

The 1254 children were evenly divided between boys and girls, the mean age of the children was 26.6 months (SD, 13.84), and the age range was 12–60 months. The majority (64%) were under two years (<24 months). Almost all (96.5%) were fully immunized, 60% had a history of illness in the past six months, 58.9% were tested for HIV, and 39.5% were attending preschool/daycare (Table 2).

### 3.3. Nutritional Status

Table 2 presents the nutritional status of children under five years. The weight of the children ranged from 2.5–38 kg with a mean weight of 11.8 kg (SD = 3.08). The overall height ranged from 84.9–112.0 cm with a mean of 84.9 (SD 12.0) (Table 3). We found that undernutrition was common. Based on the WAZ, WHZ, and HAZ cut off scores, 20.5% of the children were underweight, 17.2% were wasted, 35.8% were stunted, and 14% of the children were overweight. Of the children with undernutrition, 11% were severely underweight, 9.5% were severely wasted, and 22.4% were severely stunted according to the WHO cut off scores.

In terms of underweight and stunting, boys were more likely to be underweight and stunted than girls (23% vs. 17% and 38% vs. 33.4% respectively). There was no difference in the prevalence of wasting between the girls and the boys (16.2% vs. 17.9%). In terms of overweight, girls were more likely to be overweight than boys (14.8% vs. 13.2%).

In terms of age, older children (>24 months) were more likely to be underweight, (54.8% and 32.6%) and stunted (60.7% and 40.6% respectively). There was no difference in the prevalence of wasting between the two age groups (29% and 29% respectively).

### 3.4. Associated Factors of Stunting, Underweight and Wasting

In binary logistic regression, children who belonged to mothers with a high education level had reduced odds of being underweight (OR = 0.76, CI: 0.64–0.91, *p* = 0.003). Children who belonged to mothers with employment status had reduced odds of being wasted (OR = 0.53, CI: 0.33–0.79, *p* = 0.003), and who belonged to mothers with high income had reduced odds of being wasted (OR = 0.82, CI: 0.719–0.951, *p* = 0.008).

Children who belonged to mothers who fall into the older age category (>35 years) had increased odds of being wasted (OR = 1.61, CI: 1.08–2.38, *p* = 0.017), and being underweight (OR = 1.61, CI: 1.15–2.25, *p* = 0.005). Maternal age was not associated with childhood stunting. In terms of maternal income and childhood indicators of undernutrition, maternal income reduced the risk of stunting (OR= 0.76, CI: 0.59–0.97, *p* = 0.028) and being underweight (OR = 0.62, CI: 0.48–0.81, *p =* 0.000).

Underweight and stunting varied greatly by age, and children over the age of 24 months were three times more likely to be underweight (OR = 3.32, CI: 2.47–4.46, *p* > 0.000) and twice more likely to be stunted (OR = 1.86, CI: 1.42–2.43), *p* > 0.000). Child age was not associated with wasting. In terms of underweight by gender, male children were more likely to be underweight (OR = 1.42, CI 1.06–1.89, *p* = 0.017). There was no association between the gender of the child and the other indicators for childhood undernutrition, i.e., stunting. There was no association between the gender of the child and the other indicators for childhood undernutrition.

In terms of underweight, wasting and stunting by birth weight, high (>3.5kg) birthweight reduced the odds of childhood wasting (OR = 0.67, CI: 0.47–0.94, *p* = 0.022), and being underweight (OR = 0.73, CI: 0.55–0.97, *p* = 0.036). There was no association between birthweight and stunting.

In terms of underweight and attending daycare or preschool, children who stay at home had reduced odds of being underweight (OR = 0.64, CI: 0.48–0.88, *p* = 0.006) and reduced odds of stunting (OR = 0.68, CI: 0.52–0.90, *p* = 0.007) than those attending daycare or preschool.

Attending daycare or preschool varied by age, and children over 24 months were three times more likely to be attending daycare or preschool (2.90, CI: 2.27–3.69, *p* > 0.000). We also found that attending daycare or preschool increased the odds of a child being sick (OR = 1.50, CI: 1.19–1.91, *p* = 0.001). Maternal employment status was associated with attending daycare or preschool, and children who belonged to employed mothers were more likely to be attending daycare or preschool (OR = 2.84, CI: 2.19–3.69, *p* < 0.000).

We assessed the association between maternal age and employment status, and found that older mothers were more likely to be employed, which explains the association between maternal age and undernutrition—their children are more likely to be attending daycare or preschool (OR = 1.91, CI: 1.42–2.56, *p* < 0.000).

### 3.5. Multivariate Analysis of Factors Associated with Undernutrition

Following further analysis by multivariate logistic regression on significant risk factors of underweight, stunting, and wasting, birth weight (aOR = 0.69, CI: 0.51–0.93), child age (aOR = 3.11, CI: 2.29–4.23), child gender (aOR = 1.43, CI: 1.05–1.94) and maternal education (aOR = 0.73, CI: 0.60–0.88) remain independently associated with underweight in children under five years. A high level of maternal education reduced the odds (aOR = 0.73, CI: 0.60–0.88) of underweight. Child age (aOR = 2.54, CI: 1.44–4.49) remained significantly associated with stunting, while children over 24 months were more likely to be stunted.

Attending day care remains a significant risk factor for underweight and stunting in children under five years. Children who stayed at home had reduced odds of being underweight (aOR = 0.45, CI: 0.40–0.81) and of stunting (aOR = 0.57, CI: 0.40–0.81). Birthweight remained significantly associated with wasting and underweight. High and normal birthweight reduced the odds of childhood wasting (aOR = 0.41, CI: 0.17–0.96) and underweight (aOR = 0.69, CI: 0.17–0.96) (Table 4).

## 4. Discussion

The study assessed the nutritional status of children between the ages of 12 to 60 months recruited from health facilities in a health district in South Africa. The study showed a high prevalence of stunting (35.8%), underweight (20.5%), and wasting (17.2%) among the children. Recent data from the UNICEF indicated that 10% of underweight children who are under five years of age in the developing world are severely underweight [9]. We found that 11% of the children studied were severely underweight, 9.5% severely wasted, and 22.4% severely stunted. Based on the cut off points of the WHO, the prevalence of stunting and underweight in the study area was in the serious range and wasting was in the critical range (WHO, 2006). The prevalence of undernutrition in the current study is consistent with values reported in other studies across sub-Saharan Africa [2,12,24]. Tanzania reported higher rates of stunting, underweight and wasting of 41.9%, 46.0%, and 24.7% respectively. Similarly, the rate of stunting of 44.9% reported in an earlier study in Nigeria was relatively higher than the rates of the current study [25].

The prevalence of stunting in the district is of great public health concern. The results are higher than the national prevalence of stunting among children under five years of age, which was recorded as 21% in the latest SANHANES data (2012). Although the rates of stunting in South Africa have been between 20%–30% for the past 20 years [26], there are variations between and within provinces despite the implementation of the INP to combat the prevalence of early life malnutrition (Roadmap 2013–2017). Since the stunting rate is the preferred robust indicator of chronic undernutrition or long-term food insecurity [26], the findings indicate high prevalence of chronic undernutrition among the children in the studied communities. Concerning co-occurrence of undernutrition, 38% of the children were both stunted and underweight, 10% were stunted and wasted, and 41.4% were underweight and wasted. The results highlight the link between underweight and wasting which suggest recent loss of weight.

The multiple logistic regression showed that maternal education reduces the odds of childhood underweight. Other researchers found that maternal education has a protective effect on different under-nutrition indicators in children; for example, reference [27] found that maternal education reduced the odds of the child to be wasted, stunted, and underweight, while reference [28] reported that maternal education reduced the odds for a child to be underweight and stunted. This suggests that maternal education might be an essential factor in proper infant feeding practices, particularly because educated mothers are also likely to have a better income, which is a necessary factor in food security [28]. In univariate analysis in the current study, maternal employment status and income reduced the odds of childhood stunting and wasting. Low maternal education and women’s social status need to be taken into consideration and addressed in order to reduce undernutrition in developing countries [4].

At univariate analysis, children who belonged to mothers with an age of more than 35 years old had increased odds of being wasted and underweight than their counterparts. We found that the relationship between maternal age and undernutrition is complex, since maternal age is associated with employment status. The findings suggest that employed mothers have no impact on proper infant feeding practices during the day when they are in their workplaces. Other studies have reported that maternal employment has less positive nutrition outcomes for children under five years old; for example, in Ethiopia, maternal employment status increased the odds of wasting [17,29].

A number of studies found a significant association between maternal education and level of undernutrition among children under five years old [4,25]. We found that children who belonged to mothers with a high education level had reduced odds of being underweight. In the current study, 41.8% of the underweight children and 44.2% of children categorized as wasted belonged to mothers with low maternal education status (mothers who did not completed the 12th grade). Similar results were reported in other studies [30,31]. In order to address undernutrition in developing countries, the low levels of maternal education need to be taken into consideration. It is envisaged that when the mothers and other primary caregivers of are educated about the importance of providing children with proper nutrition, this may translate into an improved nutritional status of their children [4,14].

Similar to previous findings from sub-Saharan Africa, the sex of the child was a determinant of childhood undernutrition. We found that boys were more likely to be underweight than their girl counterparts (aOR = 143, CI: 1.05–1.94, *p* = 0.022). Results from other studies indicated that undernutrition was higher in boys than in girls in all indicators for childhood undernutrition [25,32]. Reports suggest that nutritional requirements for children increase as the child grows older, and the demand for nutrition is higher in boys than in girls. Furthermore, boys are more affected by environmental stress than girls and are more likely to show the effects of undernutrition in environments with recurrent infections [32,33]. Another report suggest that boys rarely stay at home but spend their time playing outside, whereas girls are often at home with their mothers and tend to receive small feeds that are available in the home [25].

Previous studies found a progressive rise in the rate of undernutrition with an increase in age up to the age of 24 months [27,28,32,34,35]. We also found that undernutrition and wasting were significantly higher among children over 24 months compared to the younger age group. This category is at an age during which most children require complementary food rich in protein and energy because breastfeeding has ceased [32]. If proper food is not provided, they are at risk of malnutrition as they are not food secure.

Moreover, children aged 24–60 months are at the age of attending daycare or preschool and this study showed that attending daycare or preschool increased the odds of childhood underweight and stunting, and the association was significant at multivariate analysis. Younger children attending daycare are more likely to be underweight and wasted than those cared for at home because they are breastfed less while in daycare centers [20]. The current study supports this observation. We found that 40% of the children were attending daycare and preschool and, of those, 30% were under 24 months. This suggest that some of these children should have still been breast-feeding. Other studies have found an association between daycare attendance and poor nutritional status of children under five years of age in sub-Saharan Africa. They found that daycare attendees were more likely to be moderately underweight and wasted than those cared for at home [20,36].

Child illness was not associated with all the indicators for undernutrition as reported in other studies [28]. In the current study, attending daycare or preschool increased the odds of the child being sick. Two-thirds of the children were reported to being sick in the six months prior to the survey, and, of those, 44% were attending daycare or preschool. The results support previous research, which highlighted that children attending daycare are at higher risk of acquiring infectious diseases which may affect their own health [17,18,20]. Underweight among the children in the current study might be attributed to episodes of diarrhea and upper respiratory infections, which are known risk factors for malnutrition [20]. In addition, children who are ill eat poorly and this negatively affects their nutritional status [10].

It should be noted that large numbers of daycare centers and preschools from poor-resourced settings are often not formalized and regulated [20]. They are also poorly funded, as most parents cannot afford to pay fees to cater for provision of nutritional quality food and other services such as child minders. In the current study, 73% of the mothers were unemployed and lived in informal settlements, and the source of income of almost 67% was the child support grant (R360.00 or $26.00), a monthly state grant for couples with an income of less than R4 600 ($326) or single parents earning less than R2 300 ($163). Furthermore, most often informal daycare and preschool centers are overcrowded, and thus feeding supervision and practices might be poor [20]. Care practices depend on resources for care giving, which allow the care provider to put knowledge or expertise into practice to give effective care and therefore maintain good child nutrition [4]. Strategies to improve the nutritional status of children under five years of age should not be limited to the provision of nutritious food, but must also include the promotion of good care practices [12].

## 5. Conclusions

Stunting is the most prevalent form of undernutrition in the study setting comprising urban, peri-urban and informal settlement areas. The prevalence of stunting in the district is of great public health concern, as it indicates chronic undernutrition. Whereas the risks for undernutrition are multifaceted, children who attend preschool or childcare centers have an increased risk of undernutrition. The complementary role of quality childcare in preschools and daycare centers in underprivileged communities is often lacking. This needs to be improved in order to alleviate the problem of undernutrition in underprivileged communities in South Africa.

This can be achieved by extending the services of the National School Nutritional Program administered through the Department of Basic Education to all preschools and daycare centers, and if all pre-schools and daycare centers in underprivileged settings are formalized and regulated. There is evidence that a nutritional knowledge of caregivers is a necessary requirement to determine the type and quality of diet provided to the children. Therefore, there is need to develop nutrition messages to caregivers as a strategy and intervention to improve and increase their nutritional knowledge.

The study further found that child-related factors, namely birthweight, age, and gender, increased the risk of undernutrition. The risk of underweight and stunting increased with age and coincided with the time for the cessation of breast-feeding.

### Limitations

The limitations of the use of cross-sectional data in estimating prevalence of stunting and wasting are evident in the findings. Analysis was further limited by lack of data on maternal waist circumference, which is an indicator for obesity in mothers. The statistics would enable the analysis of mother-child pair in relation to underweight, wasting and stunting. The data collection tool did not include question on the demographic descriptions of daycare centers, which would allow analysis of nutritional status by daycare centers. Lastly, we recruited the mothers from well-baby clinics and there is a likelihood that children who do not access the clinics for immunization were excluded. Nevertheless, we sampled from 18 health facilities, a representative sample in the health districts.

## Figures and Tables

**Table 1 ijerph-16-02589-t001:** Socio-demographic characteristics of mothers of children under five years of age in health facilities.

Variable	All Children	Frequency	Percent
Maternal age	>25 years	363	28.9
	25–35 years	644	51.3
	<35 years	249	19.8
Marital status	Single	909	72.7
	Ever married	342	27.3
Mother’s education level	Below high school	429	34.5
	High school	568	45.7
	Tertiary	245	19.7
Mother’s employment status	Not employed	916	73.1
	Employed	337	26.9
Receives child grant	No	399	32.9
	Yes	815	67.1
Extended family household	Yes	932	75.2
	No	308	24.8
Siblings < 5 years old	Yes	824	66.2
	No	420	33.8
Type of house	Shack	273	21.8
	Brick house	978	78.2
Access to tap water	Yes	1154	91.9
	No	8.1	8.1
Access to electricity	Yes	1202	95.9
	No	51	4.1
Have toilet in the house	Yes	774	62.3
	No	468	37.7

**Table 2 ijerph-16-02589-t002:** Characteristics of mothers of children under five years of age in health facilities.

Variable	Description	Number	Percentage
Gender (*n* = 1254)	Girls	621	49.5
	Boys	633	50.5
Child age months (*n* = 1257)	12–24 months	806	64.1
	>24 months	451	35.9
Attend daycare (*n* = 1229)	Yes	486	39.5
	No	743	60.5
Fully immunized (*n* = 1214)	Yes	1172	96.5
	No	42	3.5
History of illness	Yes	750	60.1
	No	499	40
Child tested for HIV	Yes	692	58.9
	No	483	41.1
Child HIV status (*n* = 686)	Negative	679	99.0
	Positive	7	1.0
Gestational age category (*n* = 1167)	<32 weeks	23	2.0
	32–36 weeks	86	7.4
	37–42 weeks	1058	90.6
Birth weight	Medium	862	72.38
	High	178	14.95
	Low	151	12.68
Anthropometrics	Underweight	237	20.5
	Stunting	361	35.8
	Wasting	164	17.2
	Overweight	162	14.0

**Table 3 ijerph-16-02589-t003:** Binary logistic regression for factors associated with stunting, wasting, and underweight in children under two years of age.

Variables	Stunting			Underweight			Wasting		
	OR	95% CI	*p*	OR	95% CI	*p*	OR	95% CI	*p*
Gender	1.22	0.94–1.58	0.12	1.42	1.06–1.89	0.017	1.10	0.79–1.55	0.55
Child age	1.86	1.42–2.43	0.000	3.32	2.47–4.46	0.000	1.10	0.77–1.56	0.58
Attend preschool	0.68	0.52–0.90	0.007	0.64	0.47–0.88	0.006	0.85	0.60–1.22	0.40
Fully immunized	1.16	0.54–2.50	0.69	0.76	0.35–1.65	0.50	0.78	0.31–1.96	0.61
History of illness	1.072	0.82–1.39	0.60	1.26	0.93–1.70	0.12	1.30	0.91–1.85	0.14
Child tested for HIV	0.91	0.70–1.19	0.52	0.89	0.66–1.20	0.46	1.33	0.93–1.91	0.11
Birth weight	0.83	0.64–1.07	0.16	0.73	0.55–0.97	0.036	0.67	0.47–0.94	0.022
Maternal age	0.91	0.66–1.27	0.60	1.61	1.15–2.25	0.005	1.27	1.00–1.62	0.047
Marital status	0.81	0.60–1.09	0.17	0.19	0.89–1.67	0.20	1.04	0.71–1.53	0.81
Education status	0.93	0.77–1.11	0.43	0.07	0.59–0.89	0.003	0.69	0.54–0.88	0.003
Employment status	0.97	0.72–1.30	0.84	1.02	0.74–1.41	0.87	0.51	0.33–0.79	0.003
Receives child grant	0.99	0.75–1.32	0.99	0.21	0.94–1.79	0.10	1.23	0.84–1.79	0.26
Extended family household	0.78	0.57–1.05	0.11	0.15	0.71–1.31	0.83	1.34	0.88–2.06	0.16
Siblings < 5 years old	0.94	0.72–1.24	0.69	0.96	0.71–1.31	0.83	1.16	0.81–1.64	0.40
Type of house	0.81	0.59–1.10	0.18	0.81	0.59–1.10	0.18	1.41	0.90–2.19	0.12
Access to tap water	0.99	0.63–1.55	0.98	1.51	0.84–2.72	0.16	1.14	0.61–2.11	0.66

**Table 4 ijerph-16-02589-t004:** Multivariable analysis of risk factors associated with undernutrition in children under five.

Variables	Odds Ratio	*p* Value	95% CI
**Underweight**			
Birth weight	0.69	0.015	0.51–0.93
Child age	3.11	0.000	2.29–4.23
Child gender	1.43	0.022	1.05–1.94
Maternal education	0.73	0.002	0.60–0.88
**Stunting**			
Child age	2.54	0.001	1.44–4.49
**Wasting**			
Birth weight	0.41	0.040	0.17–0.96
**Attending daycare**			
Underweight	0.45	0.001	0.40–0.81
Stunting	0.57	0.002	0.40–0.81
Wasting	1.10	0.631	0.72–1.69
Child ever sick	1.33	0.054	0.99–1.80
Child age	1.05	0.000	1.04–1.06
Employment status	3.23	0.000	2.33–4.47
Maternal age	0.80	0.245	0.54–1.16

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
