# Peer review of "Attending Informal Preschools and Daycare Centers Is a Risk Factor for Underweight, Stunting and Wasting in Children under the Age of Five Years in Underprivileged Communities in South Africa"

_ijerph, 2019, doi:10.3390/ijerph16142589_

Round 1

Reviewer 1 Report

This is a well-written manuscript; however, there are some outright errors that must be corrected. 1) line 36 -- please review what you have written as the definition for wasting -- it is not correct, as you've repeated the definition for stunting. The proper definition is noted in line 52.

2) line 370 -- you've written waste but it should read "waist."

Line 75, 76 -- "Childcare as a concept has been identified as a major factor influencing the nutritional status of children under the age of five years." You list two citations -- both from Ghana. If childcare IS a major factor and worthy of the focus of this study, it seems you ought to have more support for this statement than a single country and two references.

Is recruitment at well-baby clinics a limitation of design? It seems it would perhaps skew results -- are the most undernourished children even brought to clinics for immunization?

Your points about maternal education are strong, but are dropped when you pick up the need for improved childcare centers. What about public health education regarding breastfeeding duration, child undernutrition in general? Parental knowledge will nudge better childcare. 

This work is an important bridge between public health and policy -- thank you.

Author Response

Response to reviewer comments 1-Mokgatle et al

Reviewer no 1

This is a well-written manuscript; however, there are some outright errors that must be corrected.

1)      Line 36 -- please review what you have written as the definition for wasting -- it is not correct, as you have repeated the definition for stunting. The proper definition is noted in line 52:

Response: Thanks for highlighting this error; we made correction to the definition of wasting

2)      Line 370 -- you have written waste but it should read "waist."

Response: We have replaced maternal waste with waist

3)     Line 75, 76 -- "Childcare as a concept has been identified as a major factor influencing the nutritional status of children under the age of five years." You list two citations -- both from Ghana. If childcare IS a major factor and worthy of the focus of this study, it seems you ought to have more support for this statement than a single country and two references.

Response: Thanks for this comment, we cited additional references from other countries and added a statement from The UNICEF.

4)      Is recruitment at well-baby clinics a limitation of design? It seems it would perhaps skew results -- are the most undernourished children even brought to clinics for immunization?

Response: Immunization coverage in South Africa is lower than the global rate, which suggest that pockets of children are not accessing well baby clinics. However, recruiting children from well-baby clinics was informed by the difficulty of household surveys in settings characterized by informal settlements. We have added a sentence on the limitation of the clinics under the study limitations.

5)     Your points about maternal education are strong, but are dropped when you pick up the need for improved childcare centers. What about public health education regarding breastfeeding duration, child undernutrition in general? Parental knowledge will nudge better childcare

This work is an important bridge between public health and policy -- thank you.

Reviewer 2 Report

This study is well done and presentation is great. Conclusions are well drawn. I don't find any major corrections. My suggestions/comments are minor and directed towards authors for further clarification.

1.     Page -1, Introduction, para 2; line 36

"low height for age or wasting should be corrected as low weight for height or wasting"

2.     Page 3, Methodology, para 2.3, line 125 - socio -demographic variables

Did authors collect information on maternal (parental) intake of alcohol and illicit drugs? In the study population, if alcohol and drug use is rampant, then analyzing this information will also be helpful

3.     Page 9, line 347, $26 - can authors specify, this amount is per month or per week?

Author Response

Response to reviewer comments-2. Mokgatle et al

Reviewer no 2

This study is well done and presentation is great. Conclusions are well drawn. I don't find any major corrections. My suggestions/comments are minor and directed towards authors for further clarification.

1)      Page 1, Introduction, para 2; line 36 "low height for age or wasting should be corrected as low weight for height or wasting" We made corrections to the definition of wasting.

2)      Page 3, Methodology, para 2.3, line 125 - socio -demographic variables. Did authors collect information on maternal (parental) intake of alcohol and illicit drugs? In the study population, if alcohol and drug use is rampant, then analyzing this information will also be helpful

Response: We did not collect information on alcohol and illicit drugs as part of the maternal socio-demographic variables

3)      Page 9, line 347, $26, can the authors specify, this amount is per month or per week?

Response: The child support grant ($26) is a monthly state grant for couples with an income of less than R4 600 ($326) or single parents earning less than R2 300 ($163) and those who are unemployed. We added the explanation in the relevant section of the document.